# Dynamic and Real-Time Object Detection Based on Deep Learning for Home Service Robots

**DOI:** 10.3390/s23239482

**Published:** 2023-11-28

**Authors:** Yangqing Ye, Xiaolon Ma, Xuanyi Zhou, Guanjun Bao, Weiwei Wan, Shibo Cai

**Affiliations:** 1College of Mechanical Engineering, Zhejiang University of Technology, Hangzhou 310023, China; 1112002009@zjut.edu.cn (Y.Y.); zhouxuanyi@zjut.edu.cn (X.Z.); gjbao@zjut.edu.cn (G.B.); 2College of Mechanical and Electrical Engineering, China Jiliang University, Hangzhou 310018, China; mxlslf@cjlu.edu.cn; 3Graduate School of Engineering Science, Osaka University, Suita 562-0045, Japan; wan@sys.es.osaka-u.ac.jp

**Keywords:** real-time object detection, indoor service robots, DA-Multi-DCGAN, AT-LI-YOLO

## Abstract

Home service robots operating indoors, such as inside houses and offices, require the real-time and accurate identification and location of target objects to perform service tasks efficiently. However, images captured by visual sensors while in motion states usually contain varying degrees of blurriness, presenting a significant challenge for object detection. In particular, daily life scenes contain small objects like fruits and tableware, which are often occluded, further complicating object recognition and positioning. A dynamic and real-time object detection algorithm is proposed for home service robots. This is composed of an image deblurring algorithm and an object detection algorithm. To improve the clarity of motion-blurred images, the DA-Multi-DCGAN algorithm is proposed. It comprises an embedded dynamic adjustment mechanism and a multimodal multiscale fusion structure based on robot motion and surrounding environmental information, enabling the deblurring processing of images that are captured under different motion states. Compared with DeblurGAN, DA-Multi-DCGAN had a 5.07 improvement in Peak Signal-to-Noise Ratio (PSNR) and a 0.022 improvement in Structural Similarity (SSIM). An AT-LI-YOLO method is proposed for small and occluded object detection. Based on depthwise separable convolution, this method highlights key areas and integrates salient features by embedding the attention module in the AT-Resblock to improve the sensitivity and detection precision of small objects and partially occluded objects. It also employs a lightweight network unit Lightblock to reduce the network’s parameters and computational complexity, which improves its computational efficiency. Compared with YOLOv3, the mean average precision (mAP) of AT-LI-YOLO increased by 3.19%, and the detection precision of small objects, such as apples and oranges and partially occluded objects, increased by 19.12% and 29.52%, respectively. Moreover, the model inference efficiency had a 7 ms reduction in processing time. Based on the typical home activities of older people and children, the dataset Grasp-17 was established for the training and testing of the proposed method. Using the TensorRT neural network inference engine of the developed service robot prototype, the proposed dynamic and real-time object detection algorithm required 29 ms, which meets the real-time requirement of smooth vision.

## 1. Introduction

With the development of the economy, improvement in living standards, and intensification of the aging trend of the global society, home service robots are witnessing increasing market demand. Although the most popular robots presently working in houses execute floor-cleaning tasks, home service robots are expected to do more. In particular, they are expected to play a significant role in assisting older people and children in modern families. To perform tasks such as completing housework and interacting with family members, these robots must be able to identify and locate their target objects in real-time [1], such as clothes in dressing tasks [2] and door handles in door-opening tasks [3]. In addition, service robots usually need to change their position to facilitate task execution, which results in a high probability of blurring in the images collected by the camera on these robots. Blurred images cause difficulties in subsequent object detection processing. Moreover, objects that need to be handled in a living environment include a large number of small items (such as fruits, tableware, and toys). Also, there always exists visual occlusions between objects. Furthermore, mobile robots are powered by batteries. So, they typically use small, embedded processors with limited power consumption, memory, and computing resources, which raises the problems of memory shortage and insufficient computing power. In summary, there are several prominent challenges in the development of object detection for home service robots, including motion blur [4,5], small objects and locally occluded objects [6], and limited computing resources [7].

Existing commercial off-the-shelf (COTS) solutions, such as global shutter and increasing frame rate, can reduce motion blur to some extent. However, in low-light conditions, they may cause the images collected to be too dark or to have excessive noise. Moreover, in high-contrast scenes, they may result in the loss of details in the bright and dark areas of the images. These issues make these solutions not entirely suitable for home service robot applications. In contrast, deblurring techniques have stronger compatibility and lower costs, suitable for a wider range of environments. Furthermore, these techniques can effectively reduce motion blur without changing the camera’s hardware configuration or shooting parameters. In addition, many deblurring algorithms can retain image details while processing blur, thus obtaining clear images under various lighting and contrast conditions. Therefore, deblurring technology is a very promising solution, especially suitable for applications such as home service robots that need to obtain clear images in various environmental conditions. Deblurring, also called image sharpening, is a typical operation in image processing, which includes differential methods and filtering methods. Traditional algorithms employ mathematical principles to sharpen blurred images according to their own characteristics. However, these algorithms have the disadvantages of low generalization and adaptability. In contrast, image deblurring algorithms based on deep learning have excellent sharpening effects. For example, the hierarchical polyhedral network DMPHN [8] calculated the standard performance indicators (PSNR and SSIM) on the GoPro dataset as 30.25 and 0.935, respectively. Compared with the former, the multiscale input and multiscale output MIMO-UNet+ based on a single U-Net [9], the multiscale hierarchical network MSSNet [10], and the multistage network HINet [11] based on the Half-Instance Normalization module achieved PSNR improvements of 1.25, 1.95, and 1.51 and SSIM improvements of 0.022, 0.018, and 0.024, respectively. The above algorithms have all demonstrated good performance in their adaptability to complex images in multiple scenes. However, they fail to consider the motion information of the robot’s body, which requires a complex network structure to ensure adaptability. As a result, they rely on the support of a large server to ensure the real-time performance of the calculation (for example, the processing time of DMPHN on a single NVIDIA Tesla P100 GPU is 30 ms, the inference time of MIMO-UNet+ on an NVIDIA Titan XP is 17 ms, the inference time of HINet on an NVIDIA Tesla V100 GPU is 27 ms, and the processing time of MSSNet on an NVIDIA RTX 3090 GPU is as high as 104 ms). However, service robots usually use an embedded processing platform, which cannot supply the computing resource required by the above algorithms.

After deblurring, service robots need to detect the target object in an image. Traditional object detection algorithms usually scan the entire image using a multiscale sliding window; find the Region of Interest (ROI) [12] that may exist in the image; use the Scale-Invariant Feature Transformation (SIFT) [13], the Histogram of Oriented Gradient (HOG) [14], the Deformable Part-based Model (DPM) [15], the Local Binary Pattern (LBP) [16], and other algorithms to extract the expected features of the ROI; and, finally, use Support Vector Machine (SVM) [17,18,19], iterative adaptive boosting algorithm (Adaboost) [20], or other classifiers to classify and recognize the object. Expected features are generally defined artificially, such as edge, color, shape, and size [21,22,23,24]. The traditional object detection algorithms have pretty good accuracy in fixed scenes, but the accuracy depends on artificially designed features. Therefore, these algorithms were designed empirically and have poor generalization ability. They are suitable for specific scenarios and objects, but they cannot meet the complex scenarios and multitasking requirements of service robots in living environments.

In recent years, object-detection methods based on deep learning have been extensively studied and proved to be a promising solution. In 2014, an ROI-based convolutional network approach (R-CNN) was first introduced into the field of object detection [25]. Subsequently, Fast R-CNN [26], Faster R-CNN [27], R-FCN [28], SPP-Net [29], VGGNet [30], GoogLeNet [31], and other two-stage model object detection methods had been proposed. In 2016, an end-to-end object detection algorithm (YOLO) was proposed [32], and YOLOv2 [33], YOLOv3 [34], SSD [35], and other algorithms subsequently appeared, all belonging to single-stage models. In order to detect small objects such as partially occluded bottles in unstructured scenes, various algorithms had been employed by researchers including R-CNN, Fast R-CNN, Faster R-CNN, YOLO, YOLOv2, and SSD and their precisions reached 36.8%, 38.3%, 52.1%, 22.7%, 51.8%, and 53.2%, respectively. Although the above methods realize object detection, their precisions are not high enough for applications. The main reason is that the problem of small and locally occluded target items has not yet been resolved. Addressing the problem of small and partially occluded object detection, a dynamic discarding technique based on YOLOv3 was deployed using the NVIDIA Jetson Xavier platform for an autonomous vehicle. This algorithm could complete long-distance pedestrian detection, and the detection time of each image was 41.66 ms. This technique can only run on a Jetson GPU and does not utilize TensorRT [36]. The mobile robot equipped with an NVIDIA Jetson TX2 embedded device used the YOLOv2 algorithm to detect a table (table foot) in the office setting, achieving a precision of 69% and an image detection time of 232 ms [37]. An improved Tiny-YOLO algorithm using the ROBO structure and pruning technology was applied to the Nao robot for object detection, with a mAP of 78.75% and an inference time of 172 ms for each image [38]. A picking robot embedded with the Adreno 640 platform used a Light-YOLOv3 algorithm with a multiscale context aggregation structure and a lightweight network to detect green mangoes with an F1% of 97.7 and a model inference efficiency of 62.5 ms [39]. Another picking robot equipped with a Snapdragon 865-embedded device, running on an improved YOLOv3 algorithm was used to achieve green peach detection, with a precision of 97.3% and a test time of 60.1 ms pre-image [40]. The accuracies of the above methods had been greatly improved, and the detection of small and partially occluded objects had also achieved good results. At the same time, the efficiency of model reasoning under limited computing resources had also been optimized to a certain extent. However, the inference time of these algorithms still cannot meet the processing speed requirement of 30 FPS for real-time vision. Moreover, the above object detection algorithms cannot simultaneously meet the real-time object detection requirements of service robots in complex indoor scenes.

Researchers have also combined object detection with image deblurring algorithms. A Deblur-YOLO algorithm for detecting objects from motion-blurred images executed on a single NVIDIA GTX 1080Ti GPU was developed, with a mAP of 47.5%, a model inference efficiency of 77.2 ms, and a model size of 12.9 MB [41]. Motion-blurred images were also processed using the DeblurGANv2-InceptionResNet [42], SRN [43], DeepDeblur [44], and DynamicDeblur [45] algorithms, respectively. The time consumptions were 158 ms, 379 ms, 1550 ms, and 1525 ms, respectively, and the model sizes were 233.0 MB, 86.9 MB, 47.4 MB, and 47.8 MB, respectively. Subsequently, the YOLOv3 algorithm was used to perform object detection on clear images and motion-deblurred images by using the DeblurGANv2-InceptionResNet [42], SRN [43], DeepDeblur [44], and DynamicDeblur [45] algorithms, with a mAP of 58.5%, 42.0%, 52.3%, 51.7%, and 56.0%, respectively. However, these algorithms focused on the motion deblurring performance of images and did not pay attention to the real-time performance of both deblurring and object detection. As a result, these algorithms have complex model network structures and high inference delays, making it difficult to achieve dynamic and real-time object detection for home service robots.

Therefore, this paper proposes a dynamic and real-time object detection algorithm for home service robots based on DA-Multi-DCGAN and AT-LI-YOLO, which is able to achieve deblurring on different motion-state images based on GAN through its embedded dynamic adjustment mechanism and multiscale fusion structure. It also performs real-time object detection of target objects via the YOLOv3 model with an attention module embedded in AT-Resblock and a lightweight network unit named Lightblock. While improving the detection precision of small objects and locally occluded objects, this method can well meet the real-time requirement of object detection under the condition of limited computing resources. The main innovations and contributions of this paper are as follows:(1)A DA-Multi-DCGAN with a dynamic adjustment mechanism combining PSNR, SSIM, and sharpness and a multimodal multiscale fusion structure taking robot motion and the surrounding environment information into account is proposed, which can realize the deblurring of the acquired images under different motion states and enhance the efficiency.(2)An AT-Resblock embedded with an attention module is designed, which can fully extract features such as color and texture contours of objects, highlight key areas, integrate salient features, and improve the sensitivity and precision of small and partially occluded objects. Based on a partial depthwise separable convolution that replaces the standard convolution to improve model detection speed, a lightweight network unit named Lightblock is developed, which reduces the number of network parameters and computational complexity, and improves computational efficiency.(3)We analyzed the daily manipulating tasks of older people and children, summarized 17 kinds of frequently used daily necessities, and constructed a dataset named Grasp-17 for object detection of service robots.(4)The TensorRT neural network inference engine was run on the home service robot prototype equipped with NVIDIA Jeston AGX Xavier, and the inference efficiency was improved by three times under the same model, which achieved the dynamic object detection time of 29 ms, demonstrating the real-time nature of the proposed method in service robots during motion.

The rest of this paper is organized as follows: Section 2 describes the algorithmic framework based on DA-Multi-DCGAN and AT-LI-YOLO for home service robots, followed by a description of the experimental platform and experimental design in Section 3. The experimental results are presented and analyzed in Section 4, which is further discussed in Section 5. Finally, Section 6 concludes this work in general.

## 2. Materials and Methods

### 2.1. Framework of the Proposed Method

Our approach comprises two principal structures, DA-Multi-DCGAN and AT-LI-YOLO, which are shown in Figure 1. The DA-Multi-DCGAN algorithm module calculates, filters, and sharpens each image based on different deblurring conditions, which can effectively remove motion blur from images via the multimodal and multiscale fusion structure utilizing the robot motion and surrounding environmental information. The AT-LI-YOLO algorithm module is a multiscale detection output architecture that utilizes the AT-Resblock embedded with the attention module to emphasize key regions and incorporate significant features. And it also uses the lightweight network unit named Lightblock to reduce the number of parameters and computational complexity of the network.

### 2.2. DA-Multi-DCGAN Algorithm

The deblurring process of the DA-Multi-DCGAN algorithm module is shown in Figure 2. In the first step, the motion blur is removed by the generator of Multi-DCGAN. Secondly, the image color scale is adjusted to increase the color contrast, and the metrics of image blur (sharpness), PSNR, and SSIM are calculated. Thirdly, sharpening is performed using the Laplace operator, and the metrics of image blur, PSNR, and SSIM are updated. Next, it is judged whether the updated image blur, PSNR, and SSIM are lower than the previous indices. If yes, the program returns to perform the third step; otherwise, the deblurring process is completed and an operational image for object detection is generated.

The Multi-DCGAN adopts a deep conditional adversarial neural network with a multimodal, multiscale fusion structure, as shown in Figure 3, taking into account the robot motion and surrounding environment information. The network consists of two parts: the generator G and the discriminator D. The Conv Layer, Layer Normalization, Multiply, Multi-Resblock, Multi-Conblock, and Endblock in the generator are used to infer an initial deblurred image from the robot motion and the blurred images collected by the sensors. In the Multi-Conblock, combining shallow-layer information with robot motion mitigates network degradation caused by stacking. Additionally, it alleviates the randomness in the process of generating feature graphs, thereby enhancing model performance and efficiency under similar conditions. The generator loss function, which is presented in Equation (1), enhances image contour sharpness and overall similarity by adding $Loss_{ssim}$ and $Loss_{grad}$. The initial deblurred image and the real image inferred by the generator are simultaneously inputted to the discriminator, and the scores for the two images are outputted using the sigmoid activation function in the last layer of the discriminator, whose function is expressed by Equation (2).
(1)$$Loss_{generator}=\alpha Loss_{1}+\beta Loss_{2}+\gamma Loss_{ssim}+\delta Loss_{grad}$$
where *α*, *β*, *γ*, and *δ* are the weights between the four loss terms. $Loss_{ssim}=\frac{1-\frac{\left(2\mu_{x}\mu_{y}+c_{1}\right)\left(2\sigma_{xy}+c_{2}\right)}{\left(\mu_{x}^{2}+\mu_{y}^{2}+c_{1}\right)\left(\sigma_{x}^{2}+\sigma_{y}^{2}+c_{2}\right)}}{2}$, $Loss_{1}(x,y)={\left\|x-y\right\|}_{1}$, $Loss_{2}(x,y)={\left\|x-y\right\|}_{2}^{2}$, and $Loss_{grad}(x,y)=\frac{1}{n}\sum_{i}^{n}\left[\left|\chi_{x}\left(x_{i},y_{i}\right)\right|+\left|\chi_{y}\left(x_{i},y_{i}\right)\right|\right]$. $\mu_{x}$ and $\mu_{y}$ are the means of the original image and deblurred image, respectively. $\sigma_{x}^{2}$ and $\sigma_{y}^{2}$ are the variances of the original image and the deblurred image, respectively. $\sigma_{xy}$ is the covariance of the original image and the deblurred image. $c_{1}$, and $c_{2}$ are constants, where $c_{1}={\left(k_{1}L\right)}^{2}$ and $c_{2}={\left(k_{2}L\right)}^{2}$, with *L* being the range of image pixel values, $k_{1}=0.01$, and $k_{2}=0.03$. $\chi_{x}$ and $\chi_{y}$ are the gradient components of the original image and the initial generated image in the *x* and *y* directions.
(2)$$Loss=\underset{G}{min}\underset{D}{max}\;(log(D\left(x\right))+log(1-D\left(y\right)))$$
where *x* represents the original image, y $=G(\hat{x})$, $\hat{x}$ denotes the blurred image, and y is the deblurred image generated by the generator.

### 2.3. Structure of AT-LI-YOLO

The AT-LI-YOLO network framework, shown in Figure 4, consists of three components: Feature extraction network (FEN), Detection network (DN), and Path aggregation network (PAN).

***FEN*.** The FEN extracts features from the input image using Headblock, AT-Resblock, and convolutional layers at different scales. To optimize the model for deployment on resource-constrained devices such as robots, this paper adopts partial depthwise separable convolution instead of standard convolution. This approach reduces the network parameters and computational complexity while improving detection speed. The depthwise separable convolution consists of two components: Depthwise Convolution (DW Conv), which convolves only between the channels, and Pointwise Convolution (PW Conv), which fuses features between channels of the feature map. Using depthwise separable convolution, the FEN achieves a similar feature extraction performance to that of standard convolution, but with a significantly lower computational power [40].

Moreover, to improve the detection precision of small objects and partially occluded objects, an AT-Resblock module embedded with an attention module is designed. This module enhances the contextual relationship between feature maps in the FEN, which is based on Resblock. The attention module takes the feature maps obtained from the first and second depthwise separable convolutions in the Resblock as input and convolves them separately by means of 1 × 1 convolution kernels. The resulting feature maps are then adjusted for channel numbers using Flatten and Dense Layers and converted into one-dimensional feature maps. These one-dimensional feature maps are merged using an Add Layer and then input into a Dense Layer. The Dense Layer utilizes the sigmoid activation function to output the feature map channel saliency coefficients (Att) in the range of 0~1. These coefficients are multiplied with the Resblock output feature map to obtain a feature map of local channel saliency features, which highlights the key area and integrates the saliency features, and the detection precision of small objects and locally obscured objects is significantly improved.

***PAN.*** The feature maps extracted from the FEN are fed into the PAN, and the obtained feature maps are cascaded through the upsampling and convolution operations. Then, the feature maps of each dimension are generated by the core Lightblock of the PAN. To further reduce the computational complexity of the network, Lightblock uses a branching structure based on depthwise separable convolution instead of standard convolution. This structure divides the feature map channel into two branches, performing one and three depthwise separable convolutions, respectively, before merging the two branches with the concatenation operation. This process extracts richer features through multiple convolutions while reducing the output resolution, the number of network parameters, and computational complexity.

***DN.*** The DN employs convolution and depthwise separable convolution operations on the feature maps that pass through the PAN. This process ultimately predicts an output with *n*_channel_ channels, as expressed in Equation (3). The loss function of AT-LI-YOLO is presented in Equation (4). The classification cross-entropy loss function is applied to identify the types of the target objects. The mean square error function is adopted to locate the anchor box. And the cross-entropy loss function is employed for confidence.
(3)$$n_{channel}=3(C+4+1)$$
where the number 3 represents the predetermined anchor boxes of different types used to detect every position in the image; *C* denotes the quantity of target object types; the number 4 represents the location information of an anchor box, representing the pixel coordinates of the top-left corner of the anchor box and its length and width; and the number 1 indicates the confidence level of the detected object.
(4)$$Loss=\sum_{i=0}^{S^{2}}\sum_{j=0}^{B}I_{ij}^{obj}\left(2-\omega_{i}\times h_{i}\right)\left[{\left(x_{i}^{j}-{\dot{x}}_{i}^{j}\right)}^{2}+{\left(y_{i}^{j}-{\dot{y}}_{i}^{j}\right)}^{2}\right]\;\;\;\;\;\;\;\;\;\;\;\;\;\;\;\;\;\;+\sum_{i=0}^{S^{2}}\sum_{j=0}^{B}I_{ij}^{obj}\left(2-\omega_{i}\times h_{i}\right)\left[{\left(\sqrt{\omega_{i}^{j}}-\sqrt{{\dot{\omega }}_{i}^{j}}\right)}^{2}+{\left(\sqrt{h_{i}^{j}}-\sqrt{{\dot{h}}_{i}^{j}}\right)}^{2}\right]\;\;\;\;\;\;\;\;\;\;\;\;\;\;+\sum_{i=0}^{S^{2}}\sum_{j=0}^{B}I_{ij}^{obj}\left[{\dot{C}}_{i}^{j}\log\left(C_{i}^{j}\right)+\left(1-{\dot{C}}_{i}^{j}\right)\log\;(1-C_{i}^{j})\right]\;\;\;\;\;\;\;\;\;\;\;\;\;\;\;\;\;+\sum_{i=0}^{S^{2}}\sum_{j=0}^{B}{(1-I}_{ij}^{obj})\left[{\dot{C}}_{i}^{j}\log\left(C_{i}^{j}\right)+\left(1-{\dot{C}}_{i}^{j}\right)\log\left(1-C_{i}^{j}\right)\right]\;\;\;\;\;\;\;\;\;\;\;\;\;\;\;\;\;\;\;+\sum_{i=0}^{S^{2}}I_{ij}^{obj}\sum_{c\epsilon class}\left\{\dot{{\dot{P}}_{i}\left(c\right)\log\left[P_{i}\left(c\right)\right]+\left[1-\dot{P_{i}}\left(c\right)\right]\log\left[1-P_{i}\left(c\right)\right]}\right\}$$
where $I_{ij}^{obj}$ indicates whether there is an object in the *i*-th grid and the *j*-th anchor box. If there is an object, $I_{ij}^{obj}=1$; otherwise, $I_{ij}^{obj}=0$. ${\dot{x}}_{i}^{j}$ or $x_{i}^{j}$, ${\dot{y}}_{i}^{j}$ or $y_{i}^{j}$, ${\dot{\omega }}_{i}^{j}$ or $\omega_{i}^{j}$, ${\dot{h}}_{i}^{j}$ or $h_{i}^{j}$, and ${\dot{P}}_{i}(c)$ or $P_{i}(c)$, respectively, represent the predicted and ground truth anchor box’s top-left pixel coordinates, length and width, confidence level in prediction, and predicted object type.

## 3. Experimental Section

### 3.1. Experimental Setup

As shown in Figure 1, we developed a home service robot prototype for the experiments. The computing platform of the robot is NVIDIA Jetson AGX Xavier, which has 512 NVIDIA CUDA cores, including tensor cores and an 8-core ARM v8.2 64-bit CPU. This study utilized tensorflow2.8.1 as the deep learning framework on Ubuntu20.04. The versions of CUDA, cuDNN, and TensorRT are 11.4.14, 8.4.1, and 8.4.1, respectively. The UR3 robotic arm was controlled via the NVIDIA Jetson AGX Xavier, while the tracked chassis facilitated robot movement. A MOKOSE camera was employed for visual perception, which is capable of delivering high-resolution images up to 3840 × 2160 at a maximum speed of 30 FPS. To achieve real-time processing, the image resolution utilized in the experiments was set to 416 × 416, which is enough for object detection in household environments.

### 3.2. Dataset Preparation

To facilitate the test of object detection algorithms in indoor environments, we developed the Grasp-17 dataset. This dataset was created by analyzing the manipulating tasks of five children (2–6 years old) and five elderly people (66+ years old) over one week, as shown in Figure 5. A statistical analysis revealed that there are generally 17 kinds of objects used by both groups in their daily life. Therefore, the final predicted output channel of AT-LI-YOLO was set as 3(17 + 4 + 1), where 17 represents the type number of objects to be handled. The Grasp-17 dataset consists of two parts: one includes indoor images that are publicly available online (about 700 images from public datasets [46,47]) and the other includes images collected from the experimental platform. It contains 2000 × 2 images of 17 categories of objects (including 2000 original images and 2000 blurred images), some of which are shown in Figure 6. To facilitate the experiments, the Grasp-17 dataset was divided into two groups, named Grasp-17 (original) and Grasp-17 (blurred). The former contains images acquired when the home service robot prototype is stationary and online publicly available indoor images. The latter covers all images acquired when the robot is moving, and online publicly available indoor images, which have been motion blurred using the method described in [48].

### 3.3. Design of Experiments

Three experiments of deblurring, object detection, and integration using the robot prototype were designed to verify the proposed algorithm.

Deblurring experiment. The deblurring experiment used 80% of the Grasp-17 (blurred) dataset for training and the rest for testing. The proposed method was compared with DeblurGANv2-MobileNet [42], SRN [43], and DeblurGAN [48] in terms of standard metrics (PSNR and SSIM) and time consumption.

Object detection experiment. The object detection experiment included an ablation experiment and an advanced algorithm comparison experiment.

The ablation experiment aimed to evaluate the effectiveness of the proposed attention module and lightweight module in detecting small objects and partially occluded objects and to improve inference efficiency. The experiment involved replacing the FEN and PAN modules of YOLOv3 with two improved modules—the AT-Resblock module embedded with the attention mechanism and the lightweight module, respectively. The models were compared with the YOLOv3 and AT-LI-YOLO algorithms using the Grasp-17 (original) dataset with standard performance comparisons (including detection precision of each object, mAP, and time consumption).

The advanced algorithm comparison experiment was conducted to assess the performance of the proposed object detection algorithm in terms of detection precision and model inference for 17 daily life objects. The proposed method was compared and evaluated against Faster R-CNN [26], YOLOv3 [34], SSD [35], Light-YOLOv3 [39], Fast detection [40], YOLOv4 [49], Mask R-CNN [50], DC-SPP-YOLO [51], and YOLOv8. All the methods were trained using the Grasp-17 (original) dataset.

Integration experiment. The integration experiment was conducted to verify the precision and real-time performance of the proposed method for object detection during robot motion using the developed service robot prototype. In this experiment, 80% of the Grasp-17 (blurred) dataset was used for training, and the obtained model was then used for object detection. The performance of the proposed approach was analyzed and compared with other advanced methods.

## 4. Results

### 4.1. Deblurring Performance of DA-Multi-DCGAN

The standard performance metrics, PSNR, SSIM, and inference efficiency, were employed as the index of algorithm evaluation, as summarized in Table 1. Compared with other methods, DeblurGANv2-MobileNet and our method are more suitable for motion-blurred images in real-time. In terms of the metrics PSNR and SIMM, our method was better than others. Compared with DeblurGAN, our method showed a 5.07 improvement in PSNR and a 0.022 improvement in SIMM. However, the processing speed of SRN was much slower than DeblurGANv2-MobileNet, DeblurGAN, and our method. Several deblurred images from the Grasp-17 (blurred) dataset are shown in Figure 7. Compared with Figure 7c–e, our method produced images that are closer to the clear images in Figure 7a in terms of structural similarity, and the outlines of the objects are more prominent in all magnified detailed images. In addition, the inference speeds of all methods were improved 3 times by using the TensorRT high-performance neural network inference engine.

### 4.2. Object Detection Results

To verify the effectiveness of the attention module and lightweight module for detecting small objects and locally occluded objects, and their inference efficiency, the FEN and PAN in YOLOv3 were replaced with the structures in AT-LI-YOLO, respectively. The results are shown in Figure 8. YOLO+ (PAN of YOLOv3 with Lightblock) improved the inference speed by more than 20%, while the detection precision was similar to YOLOv3. YOLO++ (YOLOv3 with the FEN of AT-Resblock embedded with the attention mechanism) improved the detection precision of apples, oranges, and partially occluded objects by more than 10%. Compared with YOLOv3, AT-LI-YOLO improved the detection precision and model inference speed. Therefore, the proposed Lightblock in this paper, as a key technique of PAN, can enhance model inference efficiency for object detection algorithms. Additionally, the AT-Resblock embedded with the attention mechanism was designed as a key technique of FEN, which can improve the precision of small and locally occluded object detection for home service robots.

Furthermore, the proposed method was compared with Faster R-CNN [26], YOLOv3 [34], SSD [35], Light-YOLOv3 [39], Fast detection [40], YOLOv4 [49], Mask R-CNN [50], DC-SPP-YOLO [51], and YOLOv8. As before, 80% of the Grasp-17 (original) data were used for training and the rest for testing. The experimental results are shown in Table 2. Our method achieved superior precision in detecting small objects and locally occluded objects; for plastic bottles, apples, and oranges, the precision was 87.50%, 82.65%, and 90.72%, respectively. The precision of detecting other small objects and locally occluded objects like cell phones, bowls, and dishes was also high. For larger objects like packages and bananas, our method achieved a precision that was higher than more than half of the precision achieved by the SOTA algorithms. Furthermore, our method achieved a mAP of 83.05%. In terms of model inference efficiency, our method had a processing time of 75 ms, and when using TensorRT, the image detection time was reduced to 23 ms. Compared with YOLOv8, our method achieved a similar mAP and an improved precision by 15.40% and 28.15% for small objects and locally occluded objects (such as apples and oranges), respectively. Additionally, utilizing the high-performance neural network inference engine TensorRT, our method was 10 ms faster than YOLOv8. Compared with YOLOV3, our method achieved an improvement of 3.19% in mAP, and increased the detection precision of small objects and locally occluded objects, such as apples and oranges, by 19.12% and 29.52%, respectively. In addition, with the TensorRT high-performance neural network inference engine, our method was 7 ms faster than YOLOv3. Thus, our method demonstrated good precision in detecting small objects and locally occluded objects, achieved a similar mAP as YOLOv8, and maintained real-time performance in terms of algorithm inference efficiency.

Figure 9 shows the AT-LI-YOLO for detecting a single object in the field of robotic vision. Under different illumination intensities, such as bright illumination in Figure 9a, normal illumination in Figure 9b, and dark environment in Figure 9c, our method accomplished object detection with high confidence for tissues of 0.82, 0.85, and 0.84, respectively. Therefore, our method performed well in detecting a single object under different illumination intensities.

Figure 10 shows the multi-object detection experiment of AT-LI-YOLO without occlusion. In Figure 10a,b, different-dimension objects could be detected with high confidence (e.g., the confidence of bolster recognition is greater than 0.7, the confidence of tissue recognition is about 0.9, and the confidence of apple recognition is 0.92). Figure 10c,d only contained small objects. The confidence of small-object detection is relatively low, at 0.69 for a cell phone, which can also provide quite useful information for the robot and be improved by combining related environmental information and social factors. This experiment demonstrated robust object detection for objects of various dimensions without occlusion, especially for small objects.

Figure 11 shows the results of multi-object detection with occlusion. From Figure 11a,b, we can see that due to a large amount of stacking and the occlusion of the apples and oranges in a small area, many missed inspections and inaccurate detection areas appeared. Compared with YOLOv3, our method has a 19.12% improvement for apple detection with a precision of 82.65%, and a 29.52% improvement for orange detection with a precision of 90.72%. It could be concluded that our method significantly improved the precision of detecting small objects and partially occluded objects.

### 4.3. Experiment of Object Detection during the Motion State of the Service Robot

Three algorithms with high model inference efficiency and better object detection results (YOLOv3, SSD, and YOLOv4) were selected from Table 2 to be compared with the algorithms proposed in this paper (including AT-LI-YOLO) for object detection during the motion state of the service robot. In this experiment, 80% of the Grasp-17 (blurred) dataset was used to train the YOLOv3, SSD, YOLOv4, and AT-LI-YOLO algorithms. The trained models were used for object detection on our home service robot prototype. For the proposed algorithm, 80% of the Grasp-17 (blurred) dataset was used to train the DA-Multi-DCGAN model, which generated Grasp-17 (deblurred) data. Finally, the trained model of AT-LI-YOLO was used for object detection using our home service robot prototype and compared with YOLOv3, SSD, and YOLOv4 algorithms in terms of precision and model inference.

The results are shown in Figure 12. Our method achieved a mAP of 80.21%. This result is 10% higher than the other four object detection algorithms, all of which achieved mAPs below 70%. Since our algorithm utilized the DA-Multi-DCGAN module to deblur images, it is more suitable for dynamic and object detection using home service robots. Moreover, the overall model achieved an inference efficiency of 29 ms, thereby addressing the real-time vision processing time requirement of less than 30 FPS for robot object detection.

## 5. Discussion

The real-time and accurate recognition and localization of objects are prerequisites for home service robots to perform daily tasks. For robotic vision, the house environment raises the challenges of small objects and obstruction objects, image blurring caused by robotic movements, and limited computing resources. To address these issues, we proposed a blurred image processing algorithm with high efficiency and precision. In this paper, the robot motion is introduced into the deblurring algorithm, using the proposed DA-Multi-DCGAN algorithm, which can effectively suppress the gradient explosion, reduce the randomness in the feature map generation process and improve the efficiency. Experiments show that the proposed algorithm can significantly improve the precision and efficiency of object detection for robots. By embedding the attention module to enhance the contextual relationship between feature maps in the feature extraction process, and obtaining the salient feature maps of local channels through the feature map channel saliency coefficient, a high-efficiency and high-precision object detection algorithm AT-LI-YOLO was also proposed to emphasize key regions, fully extract the features such as color and texture contours of objects in images, and improve the detection precision of small objects and locally occluded objects. Furthermore, by replacing standard convolution with partially depthwise separable convolution and adding a lightweight module Lightblock, the model’s time consumption is reduced to 23 ms, meeting the real-time image processing requirements.

In the process of object recognition and manipulation by the robot, physical characteristic information and models of the target, human experiential knowledge, and scene constraints will help to significantly reduce the exploring space, lower computational complexity, and improve efficiency, such as the robot motion introduced in our algorithm. In cases of difficult local obscuration, the robot arm can be utilized to remove the interfering objects to make the target visible or the robot’s position can be changed to find the best viewing angle, reduce visual obstruction, and improve discernibility. In addition, multimodal information fusion algorithms such as oral communication and laser navigation can also be employed to narrow the robot’s search range and enhance the precision and efficiency of object detection.

As previously mentioned, blurred images pose an inherent problem in home service robots’ information acquisition, which can negatively impact the algorithm’s effectiveness and performance. Therefore, it is essential to limit the robot’s motion speed, especially during image acquisition. Excessive movement speed, particularly rotation speed, will cause severe image blur that is difficult to clear through algorithms. In addition, unplanned robot movements, such as ground bumps, structural flexibility, and acceleration/deceleration shocks, can also cause camera shake, resulting in image blurs with different patterns than those caused by planned movements. This presents a new challenge for deblurring algorithms, which we plan to address in our future research.

## 6. Conclusions

In this paper, we propose a novel object detection method for home service robots in motion states with DA-Multi-DCGAN and AT-LI-YOLO, which can be mainly concluded as follows:(1)DA-Multi-DCGAN employs a dynamic adjustment mechanism to filter and sharpen images with different blurriness. The multimodal multiscale fusion structure leverages both robot motion and surrounding environment information to effectively suppress gradient explosion and reduce the randomness of the feature map generation process. Compared with DeblurGAN, our method achieved a 5.07 improvement in PSNR and a 0.022 improvement in SIMM.(2)The proposed AT-LI-YOLO algorithm enhances the precision of detecting small and partially occluded objects by emphasizing important regions and integrating salient features through the attention module embedded in the AT-Resblock. To enhance the detection speed for home service robots, a lightweight network unit Lightblock was designed to reduce the number of parameters and computational complexity of the network. Compared with YOLOv3, the mAP of AT-LI-YOLOs increased by 3.19% and the detection speed was reduced by 7 ms.(3)The Grasp-17 dataset for home service robots was established based on the habits of elderly people and children. Experiments were conducted using the TensorRT high-performance neural network inference engine in our home service robot prototype. The results demonstrate that our method achieved real-time visual processing for home service robot motion deblurring and object detection, completing the detection process in 29 ms on the NVIDIA Jetson AGX Xavier platform.

## Figures and Tables

**Figure 1 sensors-23-09482-f001:**
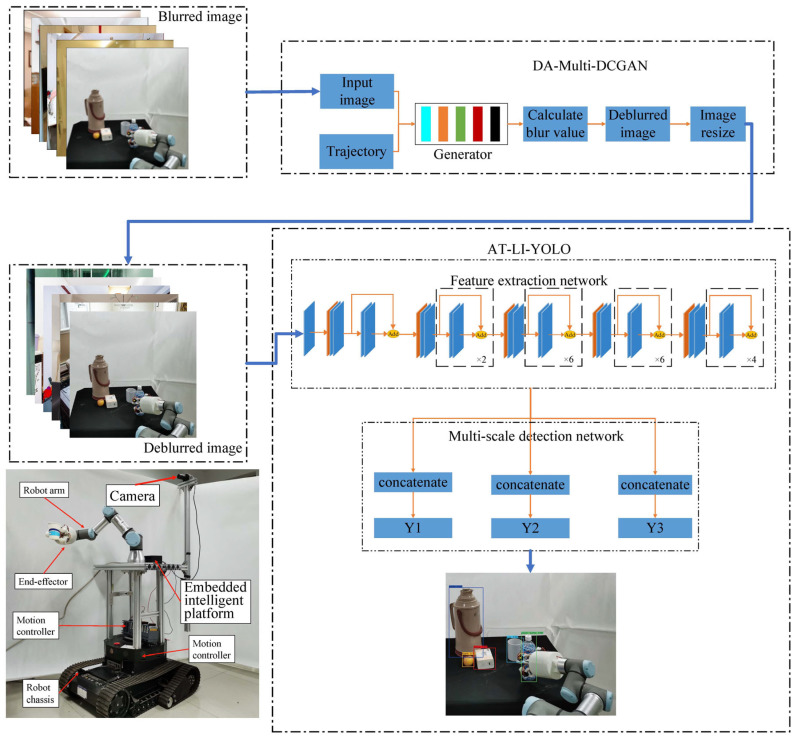
Framework of the proposed method. The Multi-DCGAN generator is fed with a blurred image and trajectory information, which is inferred during the image exposure process. The generator uses this input to produce an initial deblurred image. The image is further refined using the dynamic adjustment mechanism before being fed into the AT-LI-YOLO object detection model for detecting objects.

**Figure 2 sensors-23-09482-f002:**
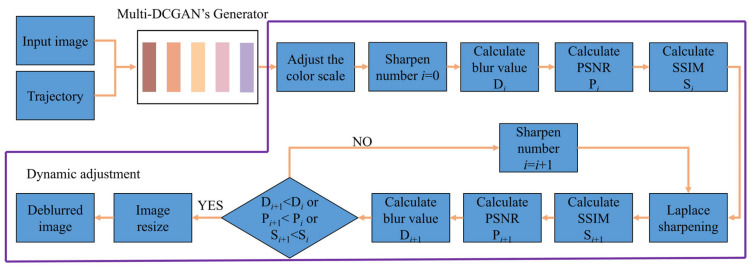
DA-Multi-DCGAN algorithm framework.

**Figure 3 sensors-23-09482-f003:**
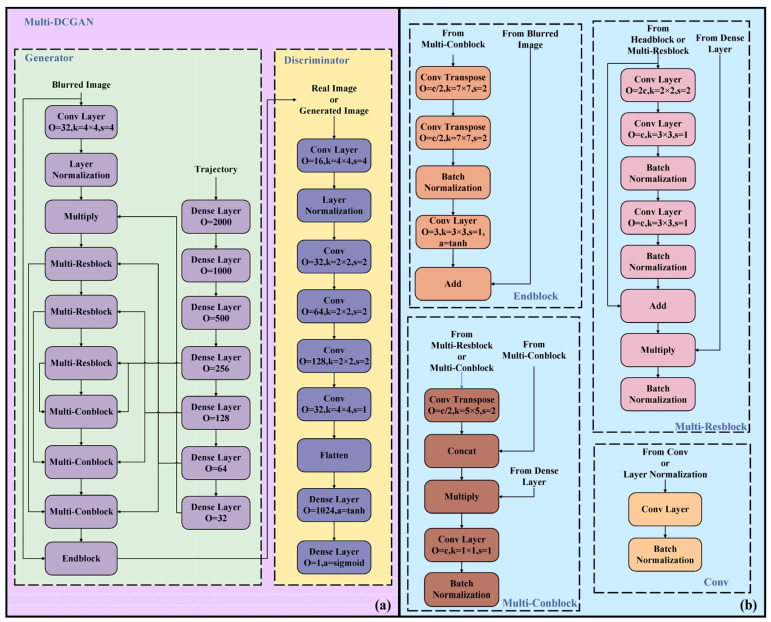
Multi-DCGAN framework. (**a**) Overall structure of Multi-DCGAN. (**b**) Specific structure of each module (O denotes the number of output channels, k denotes the convolution kernel, s denotes stride, c denotes the number of input channels, a denotes the activate function, and all activation functions use the relu function unless noted in the figure).

**Figure 4 sensors-23-09482-f004:**
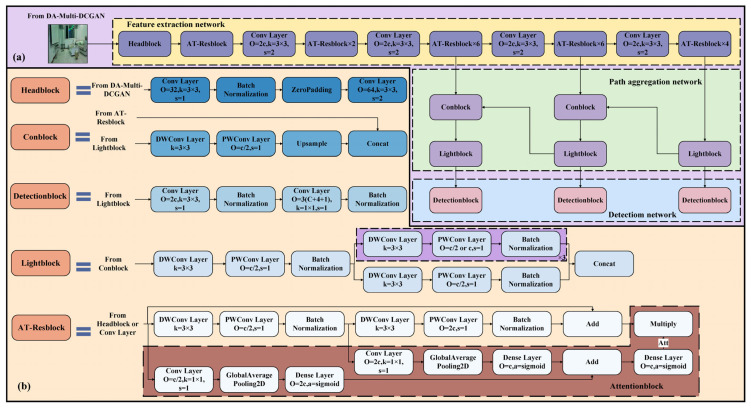
The framework of AT-LI-YOLO (**a**) Overall structure of AT-LI-YOLO; (**b**) Specific structure of each module (O denotes the number of output channels; k denotes the convolution kernel; s denotes stride; c denotes the number of the input channels; C denotes the number of types of target objects; and a denotes the activate function, with all activation function using the leaky relu function, except for the activation function indicated in the figure and the DW Conv Layer, which does not use an activation function).

**Figure 5 sensors-23-09482-f005:**
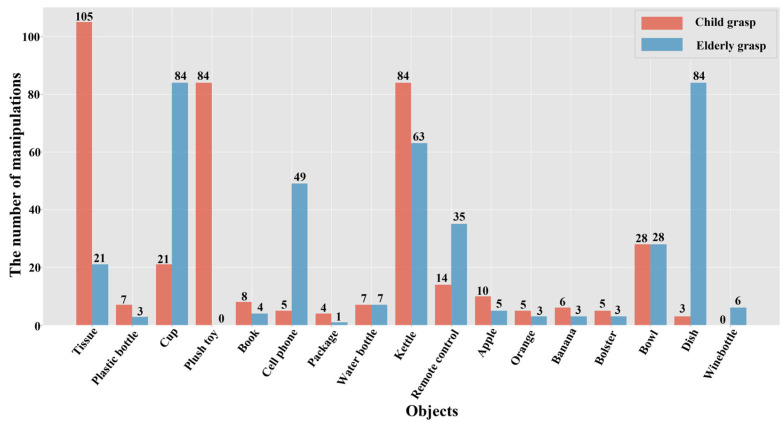
The number of times different objects are manipulated by 5 children and 5 elderly people in one week. The horizontal axis is the type of objects being manipulated, and the vertical axis is the average number of times the objects are manipulated per person per week. Red and blue represent the numbers of times these objects are manipulated by the children and elderly people in one week, respectively. Based on statistical analysis, these 17 objects that are frequently used in daily life are selected as the objects for the experiments.

**Figure 6 sensors-23-09482-f006:**
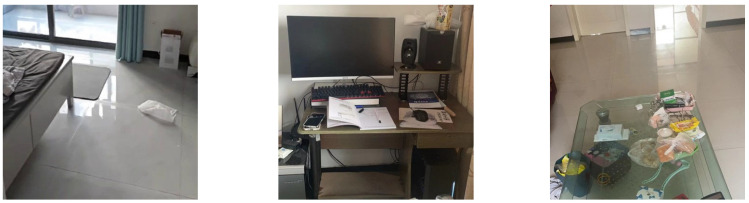
Images from Grasp-17 dataset.

**Figure 7 sensors-23-09482-f007:**
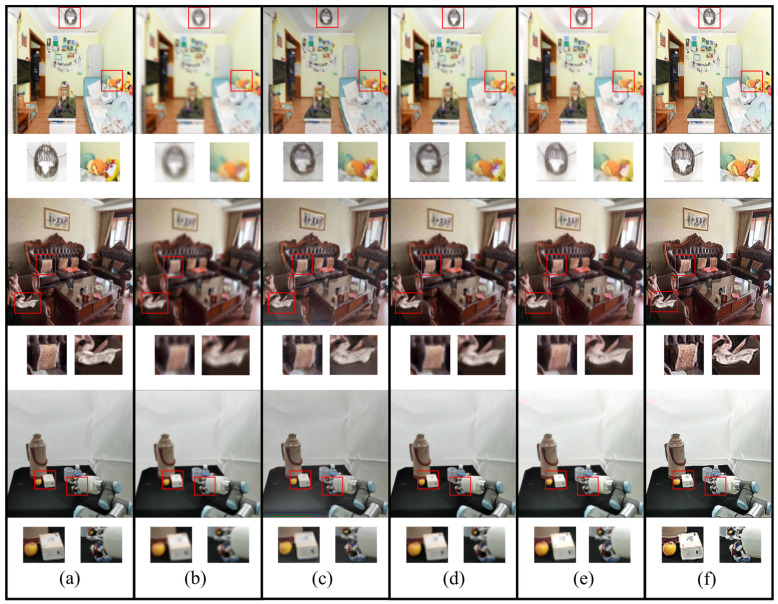
Visual comparison using the Grasp-17 (blurred). (**a**) Clear images. (**b**) Blurred images. (**c**) Deblurried images using the DeblurGAN method. (**d**) Deblurred images using the SRN method. (**e**) Deblurred images using the DeblurGANv2-MobileNet method. (**f**) Deblurred images using our method.

**Figure 8 sensors-23-09482-f008:**
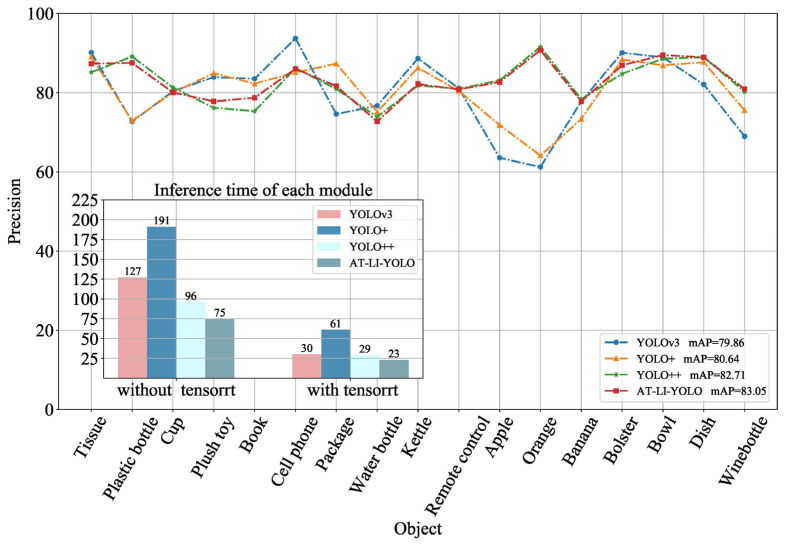
Precision of detecting different objects and inference time of different models. By replacing the FEN and PAN designed in this article from YOLOv3, the effects of different structures on detection precision and model inference efficiency were compared.

**Figure 9 sensors-23-09482-f009:**
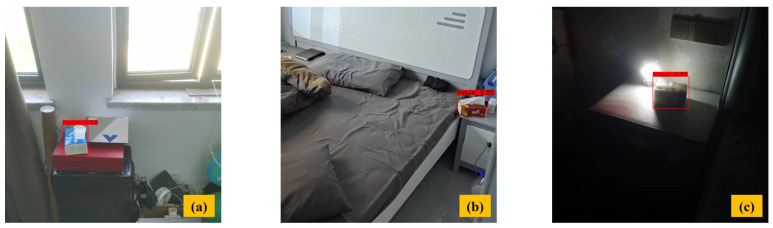
Single object detection. (**a**) Bright environment (**b**) Normal environment (**c**) Dark environment.

**Figure 10 sensors-23-09482-f010:**
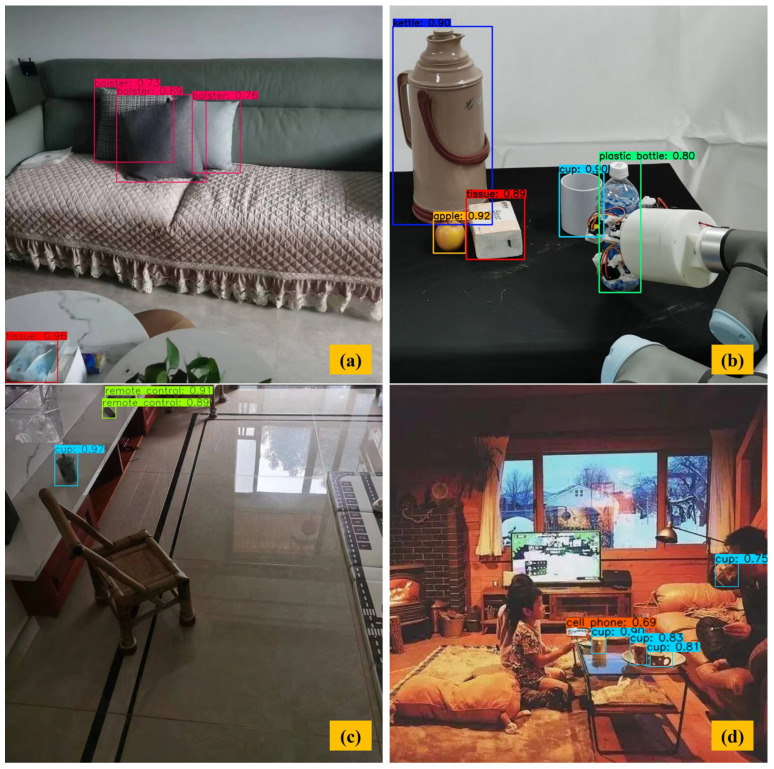
Multi-object detection without occlusion. (**a**) Including large and small objects in the image. (**b**) Including large and small objects in the image. (**c**) Including small objects in the image. (**d**) Including small objects in the image.

**Figure 11 sensors-23-09482-f011:**
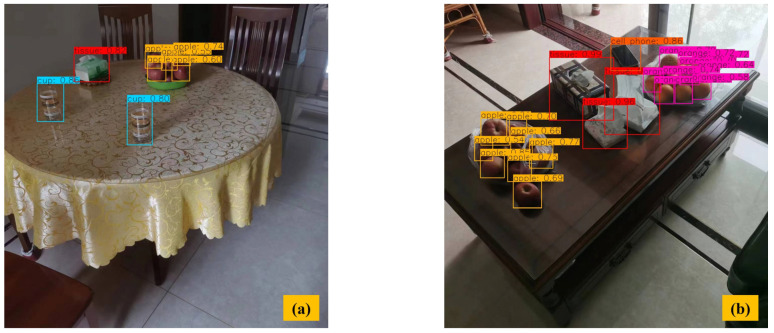
Multi-object detection with occlusion. (**a**) Single category object occlusion in image. (**b**) Multi-category object occlusion in image.

**Figure 12 sensors-23-09482-f012:**
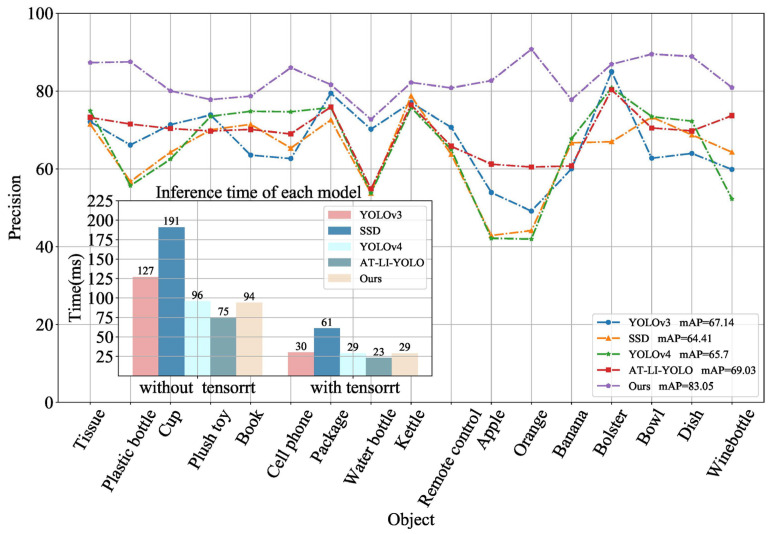
The comparison of robot object detection precision using different methods based on the Grasp-17 (blurred) dataset.

**Table 1 sensors-23-09482-t001:** Performance and efficiency comparison based on Grasp-17 (blurred).

	Original Image	Blurred Image	SRN	DeblurGAN	DeblurGANv2-MobileNet	Our Method
PSNR	-	23.932	22.170	24.852	22.940	29.920
SSIM	-	0.833	0.732	0.864	0.817	0.886
Time(ms) without TensorRT	-	-	3386	32	21	22
Time(ms) with TensorRT	-	-	996	9	6	6

**Table 2 sensors-23-09482-t002:** The comparison of object detection using different methods based on the Grasp-17 (original) dataset (%).

	SSD	Faster R-CNN	YOLOv3	Light-YOLOv3	Fast Detection	YOLOv4	Mask R-CNN	DC-SPP-YOLO	YOLOv8	AT-LI-YOLO
Tissue	80.30	89.00	90.14	80.42	80.00	82.70	62.36	73.12	91.38	87.29
Plastic bottle	56.25	69.40	72.66	64.40	68.85	52.24	64.00	62.10	79.05	87.50
Cup	83.33	73.70	80.36	65.30	75.70	68.66	63.63	74.26	81.33	80.04
Plush toy	70.00	83.70	83.90	73.90	75.20	80.37	73.48	65.70	90.67	77.77
Book	81.84	84.50	83.49	72.03	72.50	80.03	74.40	77.12	91.24	78.69
Cell phone	78.95	83.70	93.69	78.04	83.98	76.45	53.80	73.21	80.34	85.99
Package	88.00	89.30	74.58	76.66	88.31	84.42	73.10	81.46	92.25	81.63
Water bottle	75.70	74.80	76.69	72.06	75.40	57.91	76.47	74.33	78.97	72.71
Kettle	87.50	83.40	88.60	78.20	75.70	80.37	73.92	81.65	92.31	82.17
Remote control	73.60	72.40	81.10	70.06	74.20	69.38	78.94	75.82	77.00	80.81
Apple	62.40	68.70	63.53	69.30	66.48	62.30	73.30	61.48	67.25	82.65
Orange	60.80	61.10	61.20	53.70	56.40	60.50	58.70	57.69	62.57	90.72
Banana	63.10	66.80	77.78	67.10	67.79	100.00	61.74	72.30	79.32	77.77
Bolster	69.49	85.90	90.07	81.80	83.10	91.82	81.00	80.22	91.38	86.89
Bowl	90.91	77.30	88.89	80.00	83.56	76.92	60.00	67.95	83.72	89.47
Dish	93.75	62.30	82.05	75.00	80.30	85.54	65.93	79.37	88.26	88.89
Wine bottle	90.00	78.10	68.93	61.50	78.91	53.85	55.74	67.25	85.15	80.88
mAP	76.82	76.71	79.86	71.73	75.67	74.32	67.68	72.06	83.07	83.05
Time(ms) without TensorRT	191	587	127	85	95	96	604	112	131	75
Time(ms) with TensorRT	61	150	30	26	29	29	170	39	33	23

## Data Availability

The data that support the findings of this study are available from the corresponding author upon reasonable request.

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
