# Peer review of "Dynamic and Real-Time Object Detection Based on Deep Learning for Home Service Robots"

_sensors, 2023, doi:10.3390/s23239482_

Round 1

Reviewer 1 Report

Comments and Suggestions for Authors

This paper sets out real-time object detection based on deep learning for home service robots. Firstly, the English and grammar must be significantly improved, it was very difficult to follow the flow of the paper. 

The paper itself seems to focus on image blur reduction, in the main using a number of open source solutions. There are COTS solutions to motion blur reduction such as global shutters and increased frame rates etc, this isn't considered in the paper. However, there is little significance by way of moving the state of the art forward, in terms of application or in terms of Deep-Learning algorithm development. 

The methodology needs improving to provide a more real-word based solution focusing on object detection in a service robot context, which is currently lacking. 

Comments on the Quality of English Language

The English structure and grammar of this paper needs significant improvement. The flow, words and sentence meaning are difficult to follow and require re-reading to gain the meaning or inference of a sentence. 

Reviewer 2 Report

Comments and Suggestions for Authors

Targeting to the home service robotic applications, the paper presents a deep learning method to detect objects in real-time and dynamic environment. It first exploits the DA-Multi-DCGAN algorithm to calculate, filter and sharpen each image, which can effectively remove motion blur from images caused by robot motions. It then utilizes the AT-LI-YOLO technique to reduce the number of parameters and computational complexity of the network. The paper looks interesting, however there are still some concerns the authors may consider to improve quality of the manuscript as follows.

·         The authors may be aware that YOLOv8 is now available, why did they propose to use YOLOv3 in their algorithm? Can their algorithm be extended to YOLOv8? If yes, how? Please provide discussion about it.

·         The authors mentioned that they exploited the ‘publicly available online’ dataset in their experiment. They need to provide a reference for that dataset to show its validity. The authors should also describe how much the ‘publicly available online’ dataset was utilized in their work.

·         In the fourth paragraph of Introduction, the authors stated that

Addressing the small object and partially occluded object detection, a dynamic discarding technique based on YOLOv3 was deployed on the NVIDIA Jetson Xavier platform of an autonomous vehicle. This algorithm could complete long-distance pedestrian detection, and the detection time of each image was 41.66ms [36].”

And “However, the inference time of these algorithms still cannot meet the processing speed requirement of 30 FPS for real-time vision. Moreover, the above object detection algorithms cannot simultaneously meet the real-time object detection requirements of service robots in complex indoor scenes.”

Clearly in the reference [36], those authors did not use TensorRT in their work. In the current manuscript, as demonstrated in Figure 12, without using TensorRT, detection time of the proposed algorithm was 94ms, which is much longer than that in [36]. The authors should provide further evidence to say that their proposed approach is practical.

·         In Table 2, please highlight the results obtained by the proposed method to demonstrate its outperformance as compared with the SOTA algorithms.

Round 2

Reviewer 2 Report

Comments and Suggestions for Authors

The reviewer has no further comment.